# Extracellular Vesicle-Based Method for Detecting *MYCN* Amplification Status of Pediatric Neuroblastoma

**DOI:** 10.3390/cancers14112627

**Published:** 2022-05-26

**Authors:** Jirawan Panachan, Napat Rojsirikulchai, Nutkridta Pongsakul, Ladawan Khowawisetsut, Pongpak Pongphitcha, Teerapong Siriboonpiputtana, Takol Chareonsirisuthigul, Pitichai Phornsarayuth, Nisakorn Klinkulab, Natini Jinawath, Wararat Chiangjong, Usanarat Anurathapan, Kovit Pattanapanyasat, Suradej Hongeng, Somchai Chutipongtanate

**Affiliations:** 1Department of Pediatrics, Division of Hematology and Oncology, Faculty of Medicine Ramathibodi Hospital, Mahidol University, Bangkok 10400, Thailand; jirawan.pan@mahidol.ac.th (J.P.); pongpak.cha@mahidol.ac.th (P.P.); usanarat.anu@mahidol.ac.th (U.A.); 2Faculty of Medicine Ramathibodi Hospital, Mahidol University, Bangkok 10400, Thailand; napat.roj@student.mahidol.ac.th; 3Pediatric Translational Research Unit, Department of Pediatrics, Faculty of Medicine Ramathibodi Hospital, Mahidol University, Bangkok 10400, Thailand; nutkridta.pon@mahidol.ac.th (N.P.); wararat.chi@mahidol.ac.th (W.C.); 4Department of Parasitology, Faculty of Medicine Siriraj Hospital, Mahidol University, Bangkok 10700, Thailand; ladawan.kho@mahidol.edu; 5Department of Pathology, Faculty of Medicine Ramathibodi Hospital, Mahidol University, Bangkok 10400, Thailand; teerapong.sir@mahidol.ac.th (T.S.); takol.cha@mahidol.ac.th (T.C.); pitichai.phr@mahidol.ac.th (P.P.); nisakorn.kli@mahidol.ac.th (N.K.); 6Program in Translational Medicine, Faculty of Medicine Ramathibodi Hospital, Bangkok 10700, Thailand; natini.jin@mahidol.ac.th; 7Center of Excellence for Microparticle and Exosome in Diseases, Research Department, Faculty of Medicine Siriraj Hospital, Mahidol University, Bangkok 10700, Thailand; kovit.pat@mahidol.ac.th; 8Department of Environmental and Public Health Sciences, Division of Epidemiology, University of Cincinnati College of Medicine, Cincinnati, OH 45267, USA

**Keywords:** exosomes, extracellular vesicles, liquid biopsy, microvesicles, mRNA, *MYCN*, neuroblastoma, plasma

## Abstract

**Simple Summary:**

*MYCN* gene amplification, the strongest prognostic marker of aggressive neuroblastoma, is detected on invasive biopsy tissues. This study aimed to establish a less invasive method to detect *MYCN* status based on *MYCN* mRNA contents in large extracellular vesicles or microvesicles. *MYCN* mRNA-containing microvesicles were detectable in three distinct *MYCN*-amplified neuroblastoma cell lines but absent in three neuroblastoma cells with *MYCN*-non-amplification. The feasibility of this EV-based workflow was successfully demonstrated by using the simulated samples (prepared by pulsing neuroblastoma MVs into the normal human serum) and bone marrow plasma specimens obtained from nine patients at various disease stages. Taken together, this study established the novel EV-based method for detecting *MYCN* status in pediatric neuroblastoma.

**Abstract:**

*MYCN* amplification is the strongest predictor of high-risk neuroblastoma (NB). The standard procedure to detect *MYCN* status requires invasive procedures. Extracellular vesicles (EVs) contain molecular signatures of originated cells, present in biofluids, and serve as an invaluable source for cancer liquid biopsies. This study aimed to establish an EV-based method to detect the *MYCN* status of NB. Two EV subtypes, i.e., microvesicles (MVs) and exosomes, were sequentially isolated from the culture supernatant by step-wise centrifugation, ultrafiltration, and size-exclusion chromatography. Quantitative RT-PCR was performed to detect *MYCN* mRNA. As a result, *MYCN* mRNA was detectable in the MVs, but not exosomes, of *MYCN*-amplified NB cells. *MYCN* mRNA-containing MVs (*MYCN*-MV) were successfully detected in three distinct *MYCN*-amplified NB cell lines but absent in three *MYCN* non-amplification cells. The simulated samples were prepared by pulsing MVs into human serum. *MYCN*–MV detection in the simulated samples showed a less interfering effect from the human blood matrix. Validation using clinical specimens (2 mL bone marrow plasma) obtained from patients at various disease stages showed a promising result. Five out of six specimens of *MYCN*-amplified patients showed positive results, while there were no false positives in four plasma samples of the MYCN non-amplification group. This study communicated a novel EV-based method for detecting the *MYCN* status of pediatric NB based on *MYCN* mRNA contents in MVs. Future studies should be pursued in a prospective cohort to determine its true diagnostic performance.

## 1. Introduction

Neuroblastoma (NB) is the most common extracranial solid tumor in children with clinical heterogeneity [1,2,3]. NB tumor develops from the dysregulation of immature nerve cells along the sympathetic nervous system, affects 6–10% of all childhood cancers and accounts for 12–15% of all childhood cancer-related deaths [2,3,4]. Clinical presentations range from spontaneous regression to refractory/relapsed disease despite intensive therapy [2,4]. Up to 50% of patients at diagnosis have metastases to the bone marrow, bone, liver and lymph nodes [5,6]. This heterogeneity is associated with the patient’s prognosis, so that treatment for NB is based on the risk groups [7,8,9]. The International Neuroblastoma Risk Group (INRG) classification system is currently being used as a consensus approach for pretreatment risk classification by the clinical features, tumor histology and molecular markers, particularly *MYCN* status [7,8,9].

*MYCN* gene amplification (*MYCN*-amp) is the strongest prognostic marker for the high-risk aggressive NB phenotype [9,10]. *MYCN* is a proto-oncogene encoding n-Myc located on the short arm of chromosome 2 (band 2p24) and plays crucial roles in the human fetal brain during neurogenesis [10,11]. *MYCN* dysregulation is observed in NB and other neurological malignancies such as glioblastoma, retinoblastoma, medulloblastoma and ependymoma [10,11]. In NB, *MYCN*-amp is associated with other adverse prognostic factors, including unfavorable histology, mitosis karyorrhexis index and a diploid/tetraploid tumor [12]. The INRG staging system implicates MYCN-amp as the criteria of high-risk neuroblastoma that requires aggressive treatment [7,8,9]. In clinical settings, tumor *MYCN* status is detected by fluorescence in situ hybridization (FISH) on malignant tissues from tumor biopsies or metastatic cells via bone marrow aspiration/biopsy. While FISH is the gold standard method to detect *MYCN* status, this technique involves the subjective evaluation of large cell populations under a fluorescent microscope [13,14]. Tumor biopsy is an invasive procedure which sometimes results in inadequate samples and heterogenous MYCN patterns due to an abundance of non-malignant cells [13]. Moreover, it is not uncommon that the whole process of the diagnostic procedure for detecting *MYCN* status causes a delay in treatment initiation. Thus, it is imperative to develop new methods to detect *MYCN* status with less invasiveness and a shorter turnaround time using a common instrument in the clinical pathology laboratory.

The extracellular vesicle (EV) is a promising source of cancer liquid biopsy [15,16]. EVs are lipid bilayer-encapsulated nanoparticles released from all cells and present in all kinds of biofluids. EVs are known to play crucial roles in cell-to-cell communication and physiological regulation in both healthy and disease states, some of which have therapeutic potential [16,17,18,19]. EVs contain molecular signatures of original cells including lipids, proteins and nucleic acids, and thus serve as a promising source for biomarker discovery [15,16]. EVs can be classified into three subtypes based on their biogenesis pathways and bio-physico-chemical properties [18,19,20]. Exosomes (Exo) are the small EVs (40–150 nm in diameter), which originate from the endosomal pathway and multivesicular bodies, and are enriched with common exosome markers, e.g., Alix, Tsg101, Hsp70 and tetraspanins [18,19,20]. Microvesicles (MVs) are the large EVs (150–1000 nm in diameter) produced from plasma membrane budding and thus share the membrane and cytosolic proteins of parent cells [18,19,20]. Apoptotic-derived EVs or apoptotic bodies (>1000 nm in diameter) are generated from plasma membrane blebbing during cell apoptosis [18,19,20]. Exo and MVs are the main focus of the EV-based method development for several cancers, i.e., head and neck [16], brain [21], lung [22], colon [23], bladder [24], endometrial [25], ovarian [26] and prostate [27]. To the best of our knowledge, the EV-based method has never been developed for detecting the *MYCN* status of pediatric NB.

This study aimed to develop an EV workflow to detect the *MYCN* amplification status of NB. Since *MYCN*-amp NB has a high copy number of *MYCN* gene, the transcriptional product (*MYCN* mRNA) can be upregulated in the cytoplasm. We therefore hypothesized that the high levels of *MYCN* mRNA (and the translational product n-Myc) were also transferred from cells to their EVs, and that could be a basis of an EV-based method to detect *MYCN* status in pediatric NB. Both large and small EV subtypes (MVs and Exo, respectively) were compared to elucidate the suitable EV subtype for *MYCN* mRNA detection. Three distinct cell lines of each *MYCN* genomic status were applied to assess the generalizability of the developed method, while the EV pulsing experiment was conducted to determine any interfering effect of the human blood matrix on *MYCN* mRNA-containing EV detection. Finally, a proof-of-concept study was performed to demonstrate the potential applicability of EV-based liquid biopsy workflow for detecting *MYCN* status in patient-derived bone marrow plasma specimens.

## 2. Materials and Methods

### 2.1. Cell Culture

Six NB cell lines were obtained from the American Type Culture Collection (ATCC, Manassas, VA, USA) including SK-N-BE2 (ATCC©CRL-2271^TM^), SK-N-DZ (ATCC©CRL-2149^TM^), IMR-32 (ATCC©CCL-127^TM^), SH-SY5Y (ATCC©CRL-2266^TM^), SK-N-AS (ATCC©CRL-2137^TM^), and SK-N-SH (ATCC©HTB-11^TM^). Cells were maintained in the culture media as follows: SK-N-BE2, DME/F-12 (Cytiva); SK-N-SH, DMEM (1 g/L glucose) (Cytiva, Logan, UT, USA); SK-N-DZ, SK-N-AS and SH-SY5Y, DMEM (4.5 g/L glucose) (Cytiva); IMR-32, EMEM (ATCC), supplemented with 10% exosome-depleted fetal bovine serum (A2720801; Thermo Scientific, Rockford, IL, USA), 2 mM L-glutamine, 1 × Penicillin/Streptomycin (Gibco, Grand Island, NY, USA) under 5% CO_2_ at 37 °C until reaching 80% confluence. The culture supernatant was then harvested every 48 h for EV isolation.

### 2.2. Extracellular Vesicle Isolation

One hundred milliliters of the culture supernatant were subjected to EV isolation. The supernatants were centrifuged at 2000× *g*, 4 °C for 10 min to remove cells and apoptotic bodies, and then were centrifuged at 15,000× *g*, 4 °C for 20 min to collect MV pellet. The remaining supernatants were filtered through a 0.2-µm syringe filter, concentrated by 100 K MWCO centrifugal filtration (Cytiva, Global Life Sciences Solutions USA LLC., Marlborough, MA, USA) and then subjected to Exo isolation by using the qEV size exclusion chromatography column (qEVoriginal/35 nm; Izon Science, Christchurch, New Zealand) following the manufacturer’s instructions. All Exo-containing fractions were concentrated by 10 K MWCO centrifugal filtration (Cytiva, Global Life Sciences Solutions USA LLC.) to the final volume of 100 µL.

### 2.3. Fluorescence In Situ Hybridization (FISH)

Six NB cell lines were detected for *MYCN* amplification status (more than 10 copies) using Vysis LSI N-MYC SpectrumGreen (2p24)/CEP 2 SpectrumOrange (2p11.1-q11.1) Probes (Abbott Molecular Diagnostics, Des Plaines, IL, USA) as per the manufacturer’s instruction. FISH assay was performed under the clinical laboratory environment at the Human Genetics Unit, Department of Pathology, Faculty of Medicine Ramathibodi Hospital, Mahidol University.

### 2.4. Transmission Electron Microscopy (TEM)

EVs were dropped onto formvar/carbon 200 mesh copper grids (FCF200-CU; Electron Microscopy Sciences) and incubated for 5 min at room temperature. The grid was washed with the filtrated deionized water, negatively stained using 2% uranyl acetate (#22400; Electron Microscopy Sciences) in 50% methanol for 1 min and then air-dried at room temperature. Imaging was visualized using HT 7700 transmission electron microscope (Hitachi, Krefeld, Germany) with an acceleration voltage of 100 kV at the Faculty of Tropical Medicine, Mahidol University, Thailand.

### 2.5. Nanoparticle Tracking Analysis (NTA)

The particles were quantitated using NanoSight NS300 (Malvern Instruments Ltd., Malvern, Worcestershire, UK). Diluted samples were injected into the sample chamber by the syringe pump. One-min video was recorded 5 times for each sample with the following parameters: camera: sCMOS; cell temperature: 25 °C; syringe pump speed: 30 μL/s. After capture, the videos were analyzed by NanoSight Software NTA 3.4 Build 3.4.003 according to the detection threshold: 5 and blur size and max jump distance: auto. The recommended measurement is 20–100 particles/frame.

### 2.6. Western Immunoblotting

EVs were lysed with RIPA buffer (Abcam, Inc., Cambridge, UK) containing 60 mg/mL dithiothreitol and SIGMAFAST™ Protease Inhibitor tablets (Sigma-Aldrich, St. Louis, MO, USA) and the protein amounts were quantitated using a Bradford assay. Proteins (20 µg) were separated by 12.5% SDS-PAGE and transferred onto a polyvinylidene difluoride (PVDF) membrane (Millipore Immobilon®-P PVDF Membrane, Merck KGaA, Darmstadt, Germany). After blocking with 5% skim milk, the membrane was incubated with a 1:1000 primary antibody as follows: anti-Alix (#2171; Cell Signaling, Danvers, MA, USA), anti-Hsp70 (ab79852; Abcam plc, Cambridge, UK), anti-Tsg101 (ab125011; Abcam plc.), anti-n-Myc (ab24193; Abcam plc.), or anti-calnexin (#40090; Cell Signaling), at 4 °C overnight. The probed membrane was washed and incubated with 1:1000 anti-rabbit (P0448; Dako, Glostrup, Denmark) or 1:2000 anti-mouse (P0447; Dako) as an appropriate at room temperature for 1 h. The detected protein bands were enhanced by SuperSignal West Pico chemiluminescence substrate (Thermo Scientific) and captured by G:BOX Chemi XRQ (SYNGENE, Cambridge, CB4 1TF, UK).

### 2.7. RNA Extraction

RNA was extracted from EVs using TRIzol^®^ reagent (Invitrogen, Carlsbad, CA, USA) and converted to cDNA using ImProm-II™ Reverse Transcription System (PROMEGA, Madison, WI, USA) following the manufacturer’s instructions performing on T100™ Thermal Cycler (BIO-RAD). Quantity and quality of RNA were determined at OD 260 nm and 260/280 ratio, respectively, by MULTISKAN GO (Thermo Scientific). Converted reaction (20 μL) was prepared with 4 μL of 5× Reaction Buffer, 4 μL of 25 mM MgCl_2_, 1 μL of 10 mM dNTP mix, 1 μL of Oligo(dT)15 primer, and 1 μL of Reverse Transcriptase addition to converting 1 μg RNA to cDNA adjusted volume to 9 μL with molecular water. Reverse transcription protocol was annealed at 25 °C for 10 min, the first strand was extended at 42 °C for 1 h, and the reverse transcriptase was heat-inactivated at 75 °C for 10 min.

### 2.8. Quantitative RT-PCR for Detecting MYCN mRNA

Quantitative RT-PCR reaction (20 μL) was prepared from 10 μL of QPCR Green Master Mix LRox (biotechrabbit GmbH, Hennigsdorf, Germany), 2 μL of each forward and reverse primer, and 6 μL of cDNA template. Gene expression was performed with CFX96^TM^ Real-Time System (BIO-RAD). The detection of *MYCN* was calculated from technical duplication and normalized with *GAPDH*. All primer sequences were as follows: *MYCN* forward 5′-CACAAGGCCCTCAGTACCTC-3′; *MYCN* reverse 5′-ATGACACTCTTGAGCGGACG-3′; *GAPDH* forward 5′-GAAGGTGAAGGTCGGAGTC-3′; *GAPDH* reverse 5′-GAAGATGGTGATGGGATTTC-3′. Cycling protocol was initially activated at 95 °C for 3 min, denatured at 95 °C for 15 s, annealed/extended at 65 °C (*MYCN*) or 61 °C (*GAPDH*) for 30 s with plate reading, the denaturation and annealing/extension steps were repeated 39 times and heated at 95 °C for 10 s, the curve was melted at 65 °C to 95 °C with a 0.5 °C increment for 5 s and the plate reading was carried out. PCR products were confirmed with 2% agarose gel electrophoresis at 120 V for 45 min.

### 2.9. MV-Pulsed Human Serum Preparation

The isolated MVs from 100 mL NB supernatant were resuspended in 100 μL PBS and then 2 × 10^8^ MV particles (as measured by NTA) were pulsed into 2 mL human AB serum (CORNING, Manassas, VA, USA). The MV-pulsed serum was subjected to MV isolation, RNA extraction and qRT-PCR to detect *MYCN* mRNA as aforementioned. Three simulated serum samples were generated independently as per the NB cell lines.

### 2.10. Clinical Specimens

Ten bone marrow plasma specimens (acellular component of bone marrow aspiration) of nine NB patients were obtained from the Ramathibodi Tumor Biobank, Faculty of Medicine Ramathibodi Hospital, Mahidol University, Thailand. Two milliliters of patient-derived bone marrow plasma were centrifuged at 15,000× *g*, 4 °C for 20 min. The pellet was subjected to RNA extraction and qRT-PCR to detect *MYCN* mRNA. All subjects gave their informed consents for inclusion into the study. This study was conducted in accordance with the Declaration of Helsinki, and the protocol was approved by the Human Research Ethics Committee, Faculty of Medicine Ramathibodi Hospital, Mahidol University (COA. MURA2018/58; with an approval of amendment on 4 April 2022).

### 2.11. Statistical Analysis

All data were presented as mean ± SEM of technical triplication. The data were analyzed using SPSS Statistics 18 and GraphPad prism 8. One-way ANOVA with Tukey’s HSD post hoc tests was used for statistical analysis. The statistical significance was defined as *p*-value < 0.05.

## 3. Results

### 3.1. Isolation, Characterization and MYCN mRNA Detection of Two EV Subtypes Released from Representative NB Cell Lines

To address whether *MYCN*-amp neuroblastoma secreted *MYCN* mRNA-containing EVs, two representative cell lines of *MYCN*-amp (SK-N-BE2) and *MYCN*-NA (SH-SY5Y) NB cells were propagated and maintained to collect their culture supernatant. Both MVs and Exo were isolated as described in the Methods section and then were characterized as per the International Society for Extracellular Vesicles (ISEV) guidelines [20]. Western blot analysis was performed to detect three EV markers (i.e., Alix, Tsg101 and Hsp70), a negative control marker calnexin (an ER marker), and MYCN encoding n-Myc protein in the cell lysate, MV and Exo samples of *MYCN*-amp (SK-N-BE2) and *MYCN*-NA (SH-SY5Y) NB cell lines (Figure 1a). As expected, the MVs and Exo of both cell lines showed positive EV markers (higher in Exo) but an absence of the negative control marker calnexin. Note that the n-Myc protein was upregulated in *MYCN*-amp NB cells compared to *MYCN*-NA cells, and this pattern was also observed in their EVs (Figure 1a). Particle evidence was then achieved by TEM (Figure 1b) and NTA (Figure 1c), demonstrating that EV subtypes from *MYCN*-amp and *MYCN*-NA NB cells had the cup-shaped vesicular morphology with different vesicle diameter; MVs, 168.0 ± 1.4 nm and 150.4 ± 2.3 nm; Exo, 118.2 ± 2.6 nm and 115.0 ± 1.3 nm, respectively. This protein and particle evidence confirmed the presence of MVs and Exo in the isolates.

Next, total RNAs were extracted from the MVs and Exo isolates for detecting *MYCN* mRNA by using a quantitative RT-PCR. Interestingly, *MYCN* mRNA was only detectable in the MVs, but not the Exo, of *MYCN*-amp NB cells (SK-N-BE2) (Figure 1d). Both EV subtypes of MYCN-NA NB cells (SH-SY5Y) showed the absence of *MYCN* mRNA signal as expected (Figure 1d). There is limited knowledge to explain the molecular selectivity of EV subtypes released from *MYCN*-amp NB cells. However, this phenomenon may be explained, at least in part, by the different biogenesis pathways of MVs and Exo [20]. Note that the MV isolation process (step-wise centrifugation) is simpler and faster than that of Exo (a combination of step-wise centrifugation, ultrafiltration, size-exclusion chromatography). We therefore focused on the use of *MYCN*-mRNA containing MV (*MYCN*-MV) as the EV-based liquid biopsy platform for detecting the *MYCN* status of NB.

### 3.2. MYCN-MV Detection in Multiple Neuroblastoma Cell Lines

To assess the generalizability of *MYCN*-MV detection as the liquid biopsy of *MYCN*-amp NB, six cell lines including three *MYCN*-amp (SK-N-BE2, SK-N-DZ, IMR-32) and three *MYCN*-NA (SH-SY5Y, SK-N-AS, SK-N-SH) NB cells were propagated and maintained for further experiments. As part of the quality control, fluorescence in situ hybridization was performed to detect *MYCN* amplification in those cell lines (Figure 2a). Genomic amplification of *MYCN* (as shown by multiple green signals in the nucleus) was successfully confirmed in SK-N-BE2 (>50 copies), SK-N-DZ (>50 copies) and IMR-32 (>50 copies) cells, while the normal *MYCN* copy number (two per cell) was evidenced in SH-SY5Y (two copies), SK-N-AS (two copies) and SK-N-SH (three copies) cells (Figure 2a). Note that SK-N-DZ and IMR-32 cells showed hyperdiploid (>2 orange signals of CEP2 in the nucleus) that can appear in neuroblastoma patients [7,9]. Next, as the transcriptional product of *MYCN* amplification, *MYCN* mRNA was upregulated at the cellular level in SK-N-BE2, SK-N-DZ and IMR-32 cells as compared to SH-SY5Y, SK-N-AS and SK-N-SH cells (Figure 2b). For *MYCN*-MV detection, MVs were then isolated from the culture supernatants of six NB cell lines. Consistently, *MYCN* mRNA upregulation was successfully demonstrated in three distinct *MYCN*-amp NB cell lines, but not in those with the normal *MYCN* copy number (Figure 2c). This result suggested that the high *MYCN* mRNA level in the cytoplasm of *MYCN*-amp NB cells could transfer into their MVs.

### 3.3. MYCN-MV Detection in the Simulated Serum Samples

Human blood contains interfering components for downstream EV detection, including enzymes (proteinase, ribonuclease) and nanoparticles besides EV (i.e., protein aggregates, lipoproteins) [28]. The MV pulsing experiment was then performed to determine the potential interference of the human blood matrix on MYCN-MV detection. MV of six NB cell lines were pulsed into the commercially available human serum to generate six simulated serum samples, following by MV isolation, RNA extraction and qRT-PCR for MYCN mRNA (Figure 3a). As a result, MYCN mRNA upregulation was successfully detected in all simulated samples of MYCN-amp NB, and vice versa for MYCN-NA group (Figure 3b). This finding suggested that the human blood matrix had less of an interfering effect on MCYN-MV detectability and supported the applicability of MYCN-MV detection workflow on clinical specimens.

### 3.4. MYCN-MV Detection in Clinical Specimens

To this end, the proof-of-concept study of *MYCN*-MV detection workflow was performed using ten bone marrow plasma samples (obtained from acellular component of bone marrow aspiration) of nine NB patients at various conditions. Note that two samples, NB01/58 and NB01/59, were obtained from the same patient at different timepoints corresponding differed disease stages. Two milliliters of plasma were applied to isolate patient-derived MVs, following by RNA extraction and qRT-PCR for *MYCN* mRNA detection. Table 1 summarizes the demographic and clinical data including tumor *MYCN* status and *MYCN*-MV detection, while the validation of *MYCN*-MV detection in clinical samples by using the agarose gel electrophoresis is demonstrated in Figure 4.

Five out of six bone marrow plasma samples obtained from patients with *MYCN*-amp NB showed the positive results of *MYCN*-MV detection, while no false positive detection was observed in four samples from the *MYCN*-NA group (Figure 4). The negative detection of *MYCN*-MV in the NB01/58 sample was evidenced while the patient had stable disease with ongoing treatment, and the positive result of the NB01/59 was observed when this patient developed the relapse disease (Figure 4). This finding also supports that the *MYCN* status of NB could be heterogenous during tumor progression or following treatment [29]. Taken together, this study provided proof-of-concept evidence to support further investigations of *MYCN*-MV as the EV-based method for detecting *MYCN* status in pediatric NB.

## 4. Discussion

This study established, for the first time, the EV-based workflow to detect the *MYCN* status of pediatric NB. The feasibility of this method has relied on the fact that the high copy number of *MYCN* gene increases *MYCN* mRNA transcripts in NB cells which can be transferred into their MVs but not Exo (Figure 1). After validations in multiple cell lines, the simulated serum and the patient-derived bone marrow plasma (Figure 2, Figure 3 and Figure 4), the positive detection of *MYCN* mRNA in the isolated MVs (or *MYCN*-MV) was consistent with *MYCN* gene amplification status. For clinical implication, the *MYCN*-MV detection workflow had a turnaround time (from MV isolation to qRT-PCR result interpretation) less than 24 h. This *MYCN*-MV method, as an adjunct of the standard diagnostic procedures, holds potential for the rapid determination of tumor *MYCN* status in order to immediately start the appropriate treatment according to the INRG classification risk group.

Application of the large EV subtype for liquid biopsy has several advantages. MV isolation, as a one-step process, is rapid and simple compared to the multi-step Exo isolation process. MVs are the lipid bilayer-encapsulated vesicles that protect their RNA contents from degradation by RNase [30], so that the EV-based method may be more stable and easier to standardize than the circulating-free DNA (cfDNA)-based approach for detecting *MYCN* status [13,31,32,33]. Given the fact that *MYCN*-MV also contain proteins, mRNA and non-coding RNA moieties of their originate cells, this EV-based workflow holds potential for further applications beyond the detection of the *MYCN* status. For example, integrative Omics analysis could be applied on patient-derived MVs to discover personalized therapeutic targets and/or molecularly guided FDA-approved drug repurposing [34,35,36]. The extension of the EV-based method for drug prioritization should be pursued in the future.

This study unexpectedly observed the discordance between n-Myc protein (presence) and *MYCN* mRNA (absence) in the Exo of SK-N-BE2 (*MYCN*-amp) NB cell line (Figure 1a,c). One argument was that the absence of *MYCN* mRNA in the Exo could be a technical issue due to the multiple steps and lengthy process of Exo isolation compared to that of MVs. This combinatorial method (step-wise centrifugation, ultrafiltration, and qEV size-exclusion chromatography) was successfully applied to isolate mesenchymal stem cell-derived exosomes with retained biofunctionalities in our previous study [19]. In addition, the positive *GAPDH* mRNA detection in all samples could serve as the internal verification of RNA extraction and RT-PCR reaction. Therefore, we hypothesized that the undetectability of *MYCN* mRNA in the Exo of SK-N-BE2 cells was a consequence of the complex nature of exosomal cargo sorting, which is, at least in part, regulated by the circadian clock [37], RNA-binding proteins [38], YBX1 RNA/DNA-binding multifunctional protein [39,40] and the intrinsic properties of RNAs [41,42]. Although the exosomal protein/RNA discordance was out of our focus, this phenomenon may have implications in the studies of cell-to-cell communication and neuroblastoma tumor microenvironment.

This study had several limitations. This study did not perform a decontamination step (using RNase/proteinase) of the external surface of MVs. Therefore, one could not exclude the possibility that the detectable *MYCN* mRNA might be contributed partly from outside the vesicles. As the method development study, the sample size of the prove-of-concept experiment was relatively small (ten plasma specimens from nine patients). The true performance of *MYCN*-MV detection workflow should be further evaluated in a larger prospective cohort. This *MYCN*-MV workflow utilizes basic instruments for MV isolation (a bench top centrifuge) and *MYCN* mRNA detection (qRT-PCR). However, these instruments may not be available in some settings. The MV isolation protocol (centrifugation at 15,000× *g*, 20 min) may not collect all, but rather a subset, of large EV components, as evidenced by the nanoparticle tracking analysis (Figure 1c). However, this protocol has advantages in the turnaround time and consistently detects *MYCN* mRNA from MVs of cell lines and patient specimens (Figure 2, Figure 3 and Figure 4). Nonetheless, further optimization focusing on the yield and MV populations is warranted. Since *MYCN*-MV also harbor n-Myc protein (as shown in Figure 1a), further development of an antibody-based rapid test to detect n-Myc antigen in the isolated MVs may improve the accessibility of EV-based liquid biopsy for *MYCN*-amp NB diagnosis. Last but not least, this study validated the feasibility of *MYCN*-MV detection using patient-derived bone marrow plasma. Bone marrow aspiration is part of the full work-up protocol for neuroblastoma staging. Therefore, *MYCN*-MV detection in bone marrow plasma has a potential implication as the adjunct diagnosis of MYCN-amplified neuroblastoma, providing an objective clue to support clinical decision of early treatment initiation while awaiting the official pathological report (which usually take a few day). Nonetheless, this EV-based workflow would be more appealing for clinical applications if it could be performed on peripheral blood or urine. A prospective study, collecting the paired specimens of bone marrow plasma, peripheral blood plasma and urine at the first diagnosis of neuroblastoma, should be pursued as part of the method optimization and performance evaluation of *MYCN*-MV method for detecting *MYCN* amplification status of pediatric neuroblastoma in future.

## 5. Conclusions

This study developed the EV-based workflow to identify the *MYCN* status of pediatric NB by detecting *MYCN*-MVs in patient-derived plasma specimens. Future studies should investigate the EV-based method as an adjunct to the standard procedures, assessing its diagnostic performance in the large prospective cohort and evaluating its clinical impact on the early treatment initiation in patients with *MYCN*-amplified high-risk neuroblastoma.

## Figures and Tables

**Figure 1 cancers-14-02627-f001:**
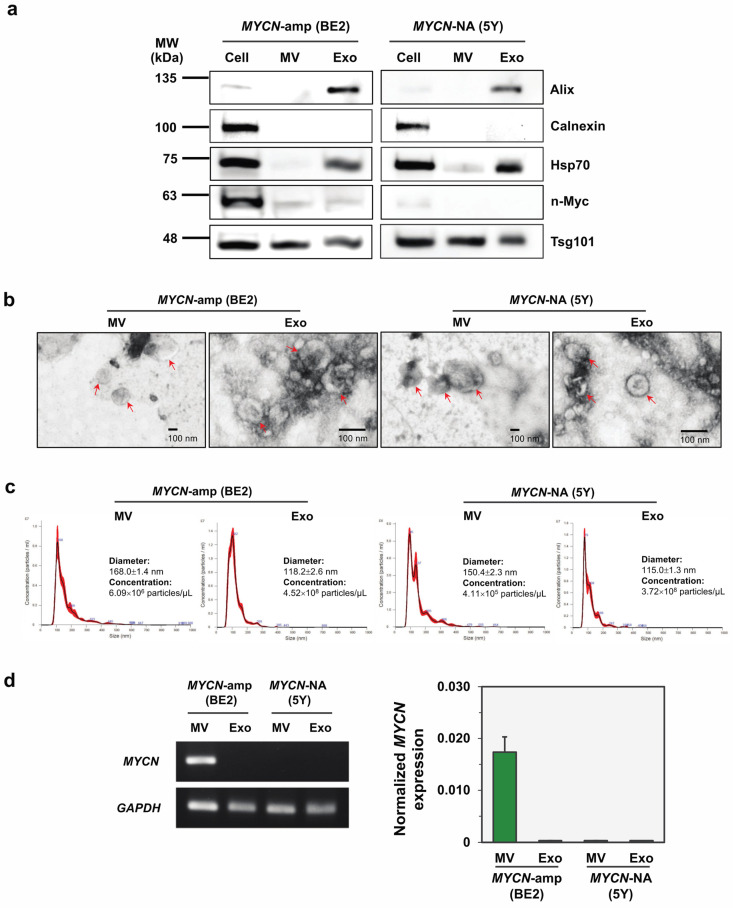
Characterization of two EV subtypes released from *MYCN*-amp and *MYCN*-NA NB cell lines. (**a**) Western blot analysis of the positive (Alix, Tsg101, Hsp70) and negative (calnexin) EV markers, and MYCN encoding n-Myc protein in cells, MVs and Exo of SK-N-BE2 and SH-SY5Y. The full-length blot images were provided in Appendix A. (**b**) Transmission electron microscopy of MVs and Exo released from *MYCN*-amp and *MYCN*-NA NB cells. The arrows indicate the nanoscale round-shaped vesicles with the size difference between MVs and Exo. Scale bar, 100 nm. (**c**) Nanoparticle tracking analysis showed the vesicle size distribution of MVs and Exo. (**d**) Quantitative RT-PCR revealed that *MYCN* mRNA could be detected in MVs, but not Exo, of *MYCN*-amp NB. The PCR products are shown on the left panel, while the *MYCN* mRNA expression (normalized to *GAPDH*) is demonstrated on the right panel. Abbreviations: 5Y, SH-SY5Y; BE2, SK-N-BE2; Exo, exosomes; MV, microvesicles; *MYCN*-amp, *MYCN* amplification; *MYCN*-NA, *MYCN* non-amplification.

**Figure 2 cancers-14-02627-f002:**
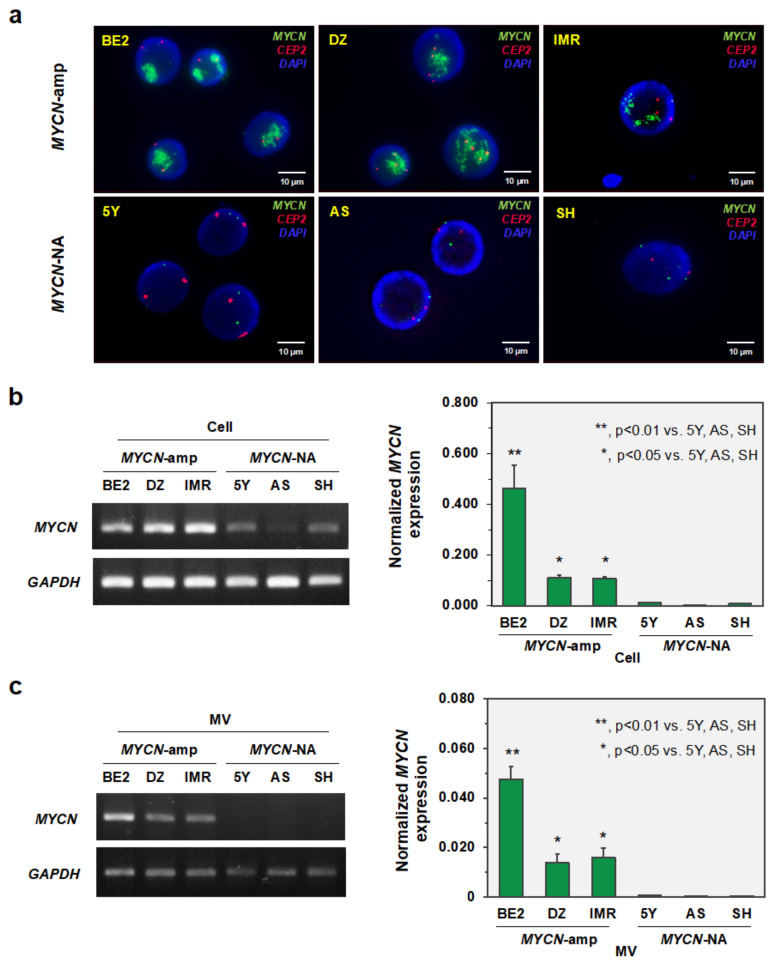
Detection of MYCN amplification status of six NB cell lines and their MVs. (**a**) Fluorescent in-situ hybridization (FISH) using the DNA probes for MYCN (2p24; green) and CEP2 (2p11.1-q11.1; orange). Quantitative RT-PCR was performed to detect MYCN mRNA in (**b**) six neuroblastoma cell lines and (**c**) their corresponding MV isolates. The PCR products were validated on agarose gel electrophoresis. MYCN mRNA expression level was normalized to GAPDH. Abbreviations: 5Y, SH-SY5Y; AS, SK-N-AS; BE2, SK-N-BE2; DZ, SK-N-DZ; IMR, IMR-32; MV, microvesicles; MYCN-amp, MYCN amplification; MYCN-NA, MYCN non-amplification; SH, SK-N-SH.

**Figure 3 cancers-14-02627-f003:**
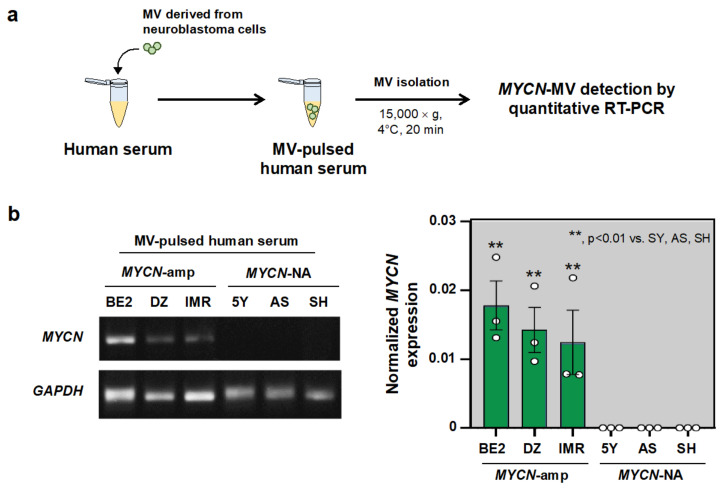
MYCN-MV detection in the simulated serum specimens. (**a**) A schematic diagram represents the workflow of simulated sample generation by pulsing MVs derived from NB cells into the human serum. Three simulated serum samples were generated independently as per the NB cell lines. Then, the simulated serum samples were subjected to MV isolation, RNA extraction and MYCN mRNA measurement by qRT-PCR. (**b**) MYCN-MV detection in six simulated samples by qRT-PCR demonstrated by the agarose gel electrophoresis and the normalized MYCN mRNA expression to GAPDH. Abbreviations: 5Y, SH-SY5Y; AS, SK-N-AS; BE2, SK-N-BE2; DZ, SK-N-DZ; IMR, IMR-32; MV, microvesicle; MYCN-amp, MYCN amplification; MYCN-MV, MYCN mRNA-containing microvesicle; MYCN-NA, MYCN non-amplification; SH, SK-N-SH.

**Figure 4 cancers-14-02627-f004:**
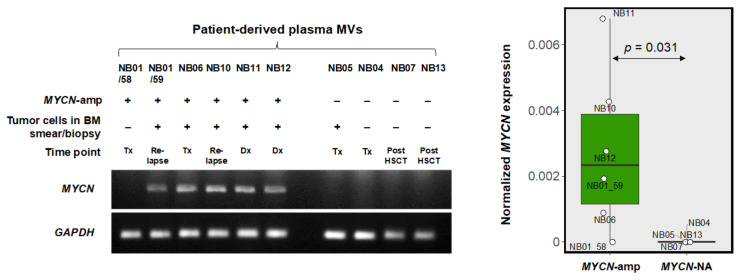
*MYCN*-MV detection in patient-derived bone marrow plasma specimens. MVs were isolated from 2 mL bone marrow plasma of NB patients and then were subjected to RNA extraction and qRT-PCR. The results are shown by the agarose gel electrophoresis (left panel) and the normalized *MYCN* mRNA expression to *GAPDH* (right panel). Abbreviations: BM, bone marrow; Dx, 1st diagnosis; HSCT, hematopoietic stem cell transplant; *MYCN*-amp, *MYCN* amplification; *MYCN*-MV, *MYCN* mRNA-containing microvesicle; *MYCN*-NA, *MYCN* non-amplification; Tx, ongoing treatment.

**Table 1 cancers-14-02627-t001:** MYCN-MV detection in 10 bone marrow plasma specimens obtained from nine neuroblastoma patients.

Sample ID	Age at First Diagnosis	Gender	INRG Stage	Tumor *MYCN* Status (Average Signal/Cell)	Timepoint of Sample Collection	Bone Marrow Smear/Biopsy	*MYCN*-MV Detection	Clinical Status at Last Visit
NB01/58	5 year	Male	M	Amplification (26.38)	Ongoing treatment	Negative for tumor	Negative	Death
NB01/59					Relapse disease	Positive for tumor	Positive	
NB06	2 year 8 month	Female	M	Amplification (>50)	Ongoing treatment	Positive for tumor	Positive	Alive without disease
NB10	10 year 5 month	Female	M	Amplification (>50)	Relapse disease	Positive for tumor	Positive	Death
NB11	1 year 3 mo	Female	M	Amplification (>50)	first diagnosis	Positive for tumor	Positive	Death
NB12	1 year	Male	M	Amplification (11.65)	first diagnosis	Positive for tumor	Positive	Alive with disease
NB04	4 year	Male	M	Non-amplification (2.00)	Ongoing treatment	Negative for tumor	Negative	Death
NB05	1 year 9 month	Female	M	Non-amplification (4.80)	Ongoing treatment	Positive for tumor	Negative	Death
NB07	6 month	Male	M	Non-amplification (2.00)	FU, post-HSCT	Negative for tumor	Negative	Alive without disease
NB13	5 year 11 month	Male	M	Non-amplification (2.00)	FU, post-HSCT	Negative for tumor	Negative	Alive with disease

Abbreviations: FU, follow-up; HSCT, hematopoietic stem cell transplant; INRG, International Neuroblastoma Risk Group; *MYCN*-MV, *MYCN* mRNA-containing microvesicle.

## Data Availability

All data are available in the main text or the Appendix A.

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
