# Peer review of "Extracellular Vesicle-Based Method for Detecting *MYCN* Amplification Status of Pediatric Neuroblastoma"

_cancers, 2022, doi:10.3390/cancers14112627_

Round 1
Reviewer 1 Report
The paper entitled “Extracellular vesicle-based liquid biopsy for detecting MYCN amplification status of pediatric neuroblastoma” by Jirawan Panachan et al., aimed to establish an extracellular vesicle-based liquid biopsy protocol to detect MYCN status in neuroblastoma patients.
The study is interesting. However, there are some major concerns on how the authors performed some experiments. In addition, the manuscript needs to be edited for scientific English language.
- NTA analysis it’s not convincing as the size measured in MV and Exos is quite superimposable. This is also evident analyzing TEM images. Usually, the size of MV is more heterogenous as analyzed by NTA and range between 100 and 1000 nm, as also stated by the authors in the introduction. The expected heterogeneity in size is not evident from the NTA analysis. The reviewer wonders if a selection of the vesicle population was not made during the preparation, especially of the MV. So, the data presented are relative only to a part of the MV population secreted by the cells. Maybe the centrifugation at 15000 x g for 20 min, is not sufficient to collect all the large EV components and they are then eliminated by the filtration with a 0.2 µm-filter.
- N-Myc protein is present in both BE2-derived MV and Exos, but amplification detected only in MV, how the authors explain this result?
- In Fig. 2a should be added the scale bar in the images.
- WB of N-Myc in cells and EV (MV and Exos) should be shown for all the NB cell lines used in the study.
- The experiment with the simulated human serum is elegant. However, the reviewer wonders why the authors used artificial serum instead of artificial plasma since the validation in patients was done with plasma and not with serum?
- The protocol used to isolate MV from human bone marrow plasma is not reported, please specify.
- When analyzing the cargo of nucleic acids inside the EV it is important to perform a decontamination step (RNase / proteinase protocol) of the external surface in order to exclude the possibility that the nucleic acids are outside instead of inside the vesicles.
Author Response
Reviewer's comments
- NTA analysis it’s not convincing as the size measured in MV and Exos is quite superimposable. This is also evident analyzing TEM images. Usually, the size of MV is more heterogenous as analyzed by NTA and range between 100 and 1000 nm, as also stated by the authors in the introduction. The expected heterogeneity in size is not evident from the NTA analysis. The reviewer wonders if a selection of the vesicle population was not made during the preparation, especially of the MV. So, the data presented are relative only to a part of the MV population secreted by the cells. Maybe the centrifugation at 15000 x g for 20 min, is not sufficient to collect all the large EV components and they are then eliminated by the filtration with a 0.2 µm-filter.
RESPONSE: We thank the Reviewer for this constructive and thoughtful comment. We agree that our MV isolation protocol may not collect all, but rather a subset, of the large EV components. This 20-min MV isolation protocol may have advantages on the turnaround time and provide the consistent results in detecting MYCN mRNA from MVs of cell lines and patient’s specimens. Nonetheless, further optimization focusing on the yield and MV populations is warranted.
Accordingly, a new statement has been added to the limitation section as follows;
Page 13, lines 410-415:
“… settings. The MV isolation protocol (centrifugation at 15,000 x g, 20 min) may not collect all, but rather a subset, of large EV components, as evidenced by the nanoparticle tracking analysis (Figure 1c). However, this protocol has advantages in the turnaround time and consistently detects MYCN mRNA from MVs of cell lines and patient specimens (Figures 2-4). Nonetheless, further optimization focusing on the yield and MV populations is warranted. Since …”
- N-Myc protein is present in both BE2-derived MV and Exos, but amplification detected only in MV, how the authors explain this result?
RESPONSE: We hypothesized that SK-N-BE2 cells probably exploit different selective sorting mechanisms for N-myc protein and MYCN mRNA. Cumulative evidence suggest that the exosomal cargo sorting has been regulated by the circadian clock [1], RNA-binding proteins [2], YBX1 RNA/DNA-binding multifunctional protein [3,4], and the intrinsic properties of RNAs [5,6]. While this study did not focus on the complex molecular machinery resulting in the discordance of protein/mRNA cargo sorting in neuroblastoma exosomes, this phenomenon may have crucial roles in cell-to-cell communication within the tumor microenvironment of aggressive neuroblastoma. This interesting research topic should be pursued in the future.
References
- Yeung CC, Dondelinger F, Schoof EM, et al. Circadian regulation of protein cargo in extracellular vesicles. Sci Adv. 2022;8(14):eabc9061. doi:10.1126/sciadv.abc9061
- Statello L, Maugeri M, Garre E, et al. Identification of RNA-binding proteins in exosomes capable of interacting with different types of RNA: RBP-facilitated transport of RNAs into exosomes. PLoS One. 2018;13(4):e0195969. doi:10.1371/journal.pone.0195969
- Shurtleff MJ, Yao J, Qin Y, et al. Broad role for YBX1 in defining the small noncoding RNA composition of exosomes. Proc Natl Acad Sci U S A. 2017;114(43):E8987-E8995. doi:10.1073/pnas.1712108114
- Suresh PS, Tsutsumi R, Venkatesh T. YBX1 at the crossroads of non-coding transcriptome, exosomal, and cytoplasmic granular signaling. Eur J Cell Biol. 2018;97(3):163-167. doi:10.1016/j.ejcb.2018.02.003
- Garcia-Martin R, Wang G, Brandão BB, et al. MicroRNA sequence codes for small extracellular vesicle release and cellular retention. Nature. 2022;601(7893):446-451. doi:10.1038/s41586-021-04234-3
- Mańka R, Janas P, Sapoń K, Janas T, Janas T. Role of RNA Motifs in RNA Interaction with Membrane Lipid Rafts: Implications for Therapeutic Applications of Exosomal RNAs. Int J Mol Sci. 2021;22(17):9416. doi:10.3390/ijms22179416
A new paragraph has been added to the discussion section as follows;
Page 13, lines 387-401:
“This study unexpectedly observed the discordance between n-Myc protein (presence) and MYCN mRNA (absence) in the Exo of SK-N-BE2 (MYCN-amp) NB cell line (Figure 1a,c). One argument was that the absence of MYCN mRNA in the Exo could be a technical issue due to the multiple steps and lengthy process of Exo isolation compared to that of MVs. This combinatorial method (step-wise centrifugation, ultrafiltration, and qEV size-exclusion chromatography) was successfully applied to isolate mesenchymal stem cell-derived exosomes with retained biofunctionalities in our previous study [19]. In addition, the positive GAPDH mRNA detection in all samples could serve as the internal verification of RNA extraction and RT-PCR reaction. Therefore, we hypothesized that the undetectability of MYCNmRNA in the Exo of SK-N-BE2 cells is a consequence of the complex nature of exosomal cargo sorting, which is, at least in part, regulated by the circadian clock [37], RNA-binding proteins [38], YBX1 RNA/DNA-binding multifunctional protein [39,40], and the intrinsic properties of RNAs [41,42]. Although the exosomal protein/RNA discordance was out of our focus, this phenomenon may have implications in the studies of cell-to-cell communication and neuroblastoma tumor microenvironment.”
- In Fig. 2a should be added the scale bar in the images.
RESPONSE: Thank you for this suggestion. The scale bar has been added to Fig. 2a already.
- WB of N-Myc in cells and EV (MV and Exos) should be shown for all the NB cell lines used in the study.
RESPONSE: We appreciate this insightful comment. Since the Exo and MVs were compared using 2 representative cell lines of MYCN-amp (SK-N-BE2) and MYCN-NA (SH-SY5Y) NB cells, we then performed western blot analysis for only these 2 cell lines, but not all cell lines used in the study.
- The experiment with the simulated human serum is elegant. However, the reviewer wonders why the authors used artificial serum instead of artificial plasma since the validation in patients was done with plasma and not with serum?
RESPONSE: We are grateful that the Reviewer finds the simulated human serum experiment elegant. In this experiment, we used human AB serum (purchased from CORNING), but not the artificial plasma, by a simple reason that the human AB serum was readily available in our lab.
- The protocol used to isolate MV from human bone marrow plasma is not reported, please specify.
RESPONSE: MV isolation from human bone marrow plasma was performed as following; 2 mL of plasma were centrifuge at 15,000 x g, 4°C for 20 min. The pellet was subjected to RNA extraction and qRT-PCR to detect MYCN mRNA. This information has been added to the methods section already (page 5, lines 228-229).
- When analyzing the cargo of nucleic acids inside the EV it is important to perform a decontamination step (RNase / proteinase protocol) of the external surface in order to exclude the possibility that the nucleic acids are outside instead of inside the vesicles.
RESPONSE: This is an important comment. We agree to the Reviewer that without decontamination step, one could not exclude the possibility that the detectable MYCN mRNA were contributed partly, if not all, from outside of the vesicles. Therefore, this important statement has been added to the limitation section of the manuscript as follows;
Page 13, lines 402-405:
“…limitations. This study did not perform a decontamination step (using RNase/proteinase) of the external surface of MVs. Therefore, one could not exclude the possibility that the detectable MYCNmRNA might be contributed partly from outside the vesicles. As…”
Reviewer 2 Report
The study investigated the possibility to use MV as liquid biopsy-based approach to detect MYCN amplification in NB patients.
The approach is of interest, however there are some issues to be further investigated.
- In the MV serum pulsing how many MV did you include. In the MM only the volume is reported. This would be relevant for a widespread MV detection workflow.
- The detection method used was the BM plasma. Since the methodology would be more widely applied using plasma or sera, did the author check whether similar results could be obtained by using sera or plasma from patients instead of BM plasma?
Line 317 grammar error
Author Response
Reviewer's comments
1. In the MV serum pulsing how many MV did you include. In the MM only the volume is reported. This would be relevant for a widespread MV detection workflow.
RESPONSE: We apologize for this overlooking. The MV-pulsed serum was performed by pulsing 2 x 108 MV particles (as measured by the nanoparticle tracking analyssi) into 2 ml human serum. This information has been added to the method section as follows;
Page 5, line 220;
“The isolated MVs from 100 ml NB supernatant were resuspended in 100 ml PBS and then 2 x 108 MV particles (as measured by NTA) were pulsed into 2 ml human AB serum (CORNING, Manassas, VA, USA).”
2. The detection method used was the BM plasma. Since the methodology would be more widely applied using plasma or sera, did the author check whether similar results could be obtained by using sera or plasma from patients instead of BM plasma?
RESPONSE: This is an important comment. We have not examined this method using plasma or sera yet. This study focused on bone marrow plasma obtained during the full work-up for neuroblastoma staging, aiming to provide the objective clue of MYCN amplification status to support clinical decision of early treatment initiation while awaiting the official pathological report (which usually take a few day). However, we agree to the Reviewer that further development and optimization of MYCN-MV detection should be pursued to examine peripheral blood plasma or serum, and perhaps urine as well.
Accordingly, new statements have been added to the discussion section as follows;
Page 13, lines 418-429:
“…diagnosis. Last but not least, this study validated the feasibility of MYCN-MV detection using patient-derived bone marrow plasma. Bone marrow aspiration is part of the full work-up protocol for neuroblastoma staging. Therefore, MYCN-MV detection in bone marrow plasma has a potential implication as the adjunct diagnosis of MYCN-amplified neuroblastoma, providing an objective clue to support clinical decision of early treatment initiation while awaiting the official pathological report (which usually take a few day). Nonetheless, this liquid biopsy workflow would be more appeal for clinical applications if it could be performed on peripheral blood or urine. A prospective study, collecting the paired specimens of bone marrow plasma, peripheral blood plasma and urine at the first diagnosis of neuroblastoma, should be pursued as part of the method optimization and performance evaluation of MYCN-MV liquid biopsy for detecting MYCN amplification status of pediatric neuroblastoma in future.”
3. Line 317 grammar error
RESPONSE: We apologize the Reviewer for this grammar error.This grammar error has been corrected as follows;
Page 9, line 321:
“As a result, MYCN mRNA upregulation was successfully detected in all simulated samples of MYCN-amp NB, and vice versa for MYCN-NA group (Figure 3b).”
Reviewer 3 Report
This is an interesting paper about the possible quick diagnosis of Nmyc
amplification (NMA) neuroblastoma from serum micro vesicles/exosomes.
Major concerns:
Authors state that this is a quicker and more reliable method than the 17 years earlier described serum free DNA using methods (Gotoh et al - correctly quoted). It is questionable as the benchside part of the method may be easier and more reliable, but not for the patients who need a BM biopsy instead of simple blood sampling. Also the bone marrow involvement should not be the criteria of simple diagnostic methods, as the presence of MYCN amplification has a stronger prognostic effect than bone marrow involvement alone.
The definition of liquid biopsy is a simple and non-invasive method of obtaining samples for further analysis to obtain diagnosis. Authors talks about plasma of bone marrow samples, which is not regarded a classical liquid biopsy method. Therefore, this method could not be regarded as a "prove-of-concept" as authors mentioned in 3.4
The exact definition of NMA amplification (more than 10 copies) is missing from the text, which makes the results less convincing. The amplification rate of each examined sample (NMA and nonNMA) should be exactly mentioned in the manuscript, helping the reader to understand the exact cut of level of this examination method.
The inadequacy of exosome for examination of MYcN amplification should be discussed more detailed, it is a technical problem (longer method for extraction) or this is the consequence of the nature of exosomes.
Also questionable whether sampling post-HSCT in Follow-up period in a alive patient with no evidence of disease (Tabl 1 last two samples) could be regarded a scientifically correct method?
Author Response
Reviewer's comments
1. Authors state that this is a quicker and more reliable method than the 17 years earlier described serum free DNA using methods (Gotoh et al - correctly quoted). It is questionable as the benchside part of the method may be easier and more reliable, but not for the patients who need a BM biopsy instead of simple blood sampling. Also the bone marrow involvement should not be the criteria of simple diagnostic methods, as the presence of MYCN amplification has a stronger prognostic effect than bone marrow involvement alone.
RESPONSE: We thanks the Reviewer’s for this thoughtful comment. Bone marrow aspiration is part of a full work-up protocol for neuroblastoma staging. From a clinical viewpoint, we foresee the potential implication of MYCN-MV detection in the bone marrow plasma as the adjunct diagnosis of MYCN-amplified neuroblastoma, providing an objective clue to support clinical decision of early treatment initiation while awaiting the official pathological report (which usually take a few day). Nonetheless, we agree to the Reviewer that the MYCN-MV detection would be more appeal for further applications if it can be performed on peripheral blood or urine. A prospective study, collecting the paired specimens of bone marrow plasma, peripheral blood plasma and urine, should be pursued as part of the method optimization and performance evaluation of MYCN-MV detection for MYCN-amplified neuroblastoma in future.
As a result, new statements have been added to the discussion section as follows;
Page 13, lines 418-429:
“…diagnosis. Last but not least, this study validated the feasibility of MYCN-MV detection using patient-derived bone marrow plasma. Bone marrow aspiration is part of the full work-up protocol for neuroblastoma staging. Therefore, MYCN-MV detection in bone marrow plasma has a potential implication as the adjunct diagnosis of MYCN-amplified neuroblastoma, providing an objective clue to support clinical decision of early treatment initiation while awaiting the official pathological report (which usually take a few day). Nonetheless, this liquid biopsy workflow would be more appeal for clinical applications if it could be performed on peripheral blood or urine. A prospective study, collecting the paired specimens of bone marrow plasma, peripheral blood plasma and urine at the first diagnosis of neuroblastoma, should be pursued as part of the method optimization and performance evaluation of MYCN-MV liquid biopsy for detecting MYCN amplification status of pediatric neuroblastoma in future.”
2. The definition of liquid biopsy is a simple and non-invasive method of obtaining samples for further analysis to obtain diagnosis. Authors talks about plasma of bone marrow samples, which is not regarded a classical liquid biopsy method. Therefore, this method could not be regarded as a "prove-of-concept" as authors mentioned in 3.4
RESPONSE: We thank the Reviewer for this comment.We aware that the definition of liquid biopsy may be varied in different contexts and settings. As indicated by the NCI Dictionary of Cancer Terms, liquid biopsy is “a test done on a sample of blood to look for cancer cells from a tumor that are circulating in the blood or for pieces of DNA from tumor cells that are in the blood” (https://www.cancer.gov/publications/dictionaries/cancer-terms/def/liquid-biopsy). Another definition coined by Dr. Catherine Alix-Panabières, states that liquid biopsy is “the analysis of tumours using biomarkers circulating in fluids such as the blood” (1). We believe that the definition of liquid biopsy is still evolving, but it would be developed upon the detection of disease biomarkers (i.e., proteins, nucleic acids, lipids, glycans, EVs, etc.) in non-invasive biofluids (e.g., saliva, urine, sweat, tear) or less-invasive biofluids (i.e., blood, bone marrow, cerebrospinal fluid, plural fluid, ascites) as compared to the surgical biopsies. We hope that the Reviewer would comprehend our decision to use the term “prove-of-concept” in the section 3.4 of the current study.
Reference
- Alix-Panabières C. The future of liquid biopsy. Nature. 2020;579(7800):S9. doi: 10.1038/d41586-020-00844-5.
3. The exact definition of NMA amplification (more than 10 copies) is missing from the text, which makes the results less convincing. The amplification rate of each examined sample (NMA and nonNMA) should be exactly mentioned in the manuscript, helping the reader to understand the exact cut of level of this examination method.
RESPONSE: This is an important comment. We apology for this overlooking. Accordingly, the definition of NMA amplification (more than 10 copies) has been added to the methods section (page 4, line 151). Also, the amplification rate of each sample has been added to Table 1.
4. The inadequacy of exosome for examination of MYcN amplification should be discussed more detailed, it is a technical problem (longer method for extraction) or this is the consequence of the nature of exosomes.
RESPONSE: We thank the Reviewer for this constructive comment. Although this exosome isolation protocol (step-wise centrifugation, ultrafiltration and size exclusion chromatography) applied in this study was multi-step and lengthy, this similar protocol was successfully applied to isolate mesenchymal stem cell exosomes with retained biofunctionalities in our previous study [1]. In addition, the positive GAPDH mRNA detection could serve as a technical verification of RNA extraction and RT-PCR reaction for each sample. Thus, we hypothesized that the undetectability of MYCN mRNA in the exosomes from MYCN-amp neuroblastoma is a consequence of the complex nature of exosomal cargo sorting machinery rather than the technical issue.
Reference
- Chutipongtanate S, Kongsomros S, Pongsakul N, et al. Anti-SARS-CoV-2 effect of extracellular vesicles released from mesenchymal stem cells. J Extracell Vesicles. 2022;11(3):e12201. doi:10.1002/jev2.12201
Accordingly, a new paragraph has been added to the discussion section as follows;
Page 13, lines 387-401:
“This study unexpectedly observed the discordance between n-Myc protein (presence) and MYCN mRNA (absence) in the Exo of SK-N-BE2 (MYCN-amp) NB cell line (Figure 1a,c). One argument was that the absence of MYCN mRNA in the Exo could be a technical issue due to the multiple steps and lengthy process of Exo isolation compared to that of MVs. This combinatorial method (step-wise centrifugation, ultrafiltration, and qEV size-exclusion chromatography) was successfully applied to isolate mesenchymal stem cell-derived exosomes with retained biofunctionalities in our previous study [19]. In addition, the positive GAPDH mRNA detection in all samples could serve as the internal verification of RNA extraction and RT-PCR reaction. Therefore, we hypothesized that the undetectability of MYCN mRNA in the Exo of SK-N-BE2 cells is a consequence of the complex nature of exosomal cargo sorting, which is, at least in part, regulated by the circadian clock [37], RNA-binding proteins [38], YBX1 RNA/DNA-binding multifunctional protein [39,40], and the intrinsic properties of RNAs [41,42]. Although the exosomal protein/RNA discordance was out of our focus, this phenomenon may have implications in the studies of cell-to-cell communication and neuroblastoma tumor microenvironment.”
5. Also questionable whether sampling post-HSCT in Follow-up period in a alive patient with no evidence of disease (Table 1 last two samples) could be regarded a scientifically correct method?
RESPONSE: We appreciate this Reviewer’s comment. The bone marrow aspiration post-HSCT was performed as part of the routine follow-up to monitor relapsed disease in neuroblastoma patients with the previous history of bone marrow metastasis.
Round 2
Reviewer 1 Report
The paper remain preliminary for some aspects.
Author Response
Reviewer’s comment
The paper remain preliminary for some aspects.
RESPONSE: We appreciate this Reviewer’s comment. We are looking forward to pursue the MYCN-MV method to evaluate the diagnostic performance in a prospective cohort of pediatric neuroblastoma in future.
Reviewer 2 Report
The Authors have addressed all my concerns
Author Response
Reviewer’s commentThe Authors have addressed all my concerns.
RESPONSE: We would like to thank the Reviewer for all constructive and insightful comments which we feel help improving the quality of the revised manuscript.
Reviewer 3 Report
The paper highly improved after revision.
However, the title and text is still highly misleading, as authors try to regard bone marrow aspiration as a liquid biopsy method. It is a false information (after reading the quoted paper of Alix-Panabières C. in Nature, too). Anywhere the authors use "liquid biopsy" it should be change to bone marrow biopsy, and similarly instead of using "plasma" they should use another less misleading term such as e.g. "acellular component of bone marrow".
Author Response
Reviewer’s comments
The paper highly improved after revision. However, the title and text is still highly misleading, as authors try to regard bone marrow aspiration as a liquid biopsy method. It is a false information (after reading the quoted paper of Alix-Panabières C. in Nature, too). Anywhere the authors use "liquid biopsy" it should be change to bone marrow biopsy, and similarly instead of using "plasma" they should use another less misleading term such as e.g. "acellular component of bone marrow".
RESPONSE: This is an important comment. As the Reviewer’s suggestion, we revised the terms “liquid biopsy” and “plasma” throughout the manuscript as follows;
Title;
Line 2:
“Extracellular vesicle-based method for detecting MYCN amplification status of pediatric neuroblastoma”
Simple Summary;
Line 33:
“…to establish a less invasive method to detect MYCNstatus …”
Lines 38-39:
“… normal human serum) and bone marrow plasma specimens obtained of nine patients …”
Abstract;
Line 44:
“… to establish an EV-based method to detect MYCNstatus …”
Line 52:
“Validation using clinical specimens (2 mL bone marrow plasma) obtained from patients …”
Line 54:
“Five out of six specimens of MYCN-amplified patients …”
Line 55:
“This study communicated a novel EV-based method for detecting MYCN status …”
Introduction;
Line 108:
“Exo and MVs are the main focus of EV-based method development for several cancers …”
Line 110:
“To the best of our knowledge, EV-based method has never been developed for detecting MYCN status of pediatric NB.”
Methods;
Lines 225-226:
“Ten bone marrow plasma specimens (acellular component of bone marrow aspiration) of nine NB patients were obtained from ....”
Results;
Line 335:
“…MYCN-MV detection workflow was performed using ten bone marrow plasma samples (obtained from acellular component of bone marrow aspiration) of nine NB patients …”
Line 347:
“Figure 4. MYCN-MV detection in patient-derived bone marrow plasma specimens. MVs were …”
Line 354:
“Five out of six bone marrow plasma obtained from patients withMYCN-amp NB …”
Discussion;
Line 369:
“… the simulated serum and the patient-derived bone marrow plasma …”
Line 379:
“… so that EV-based method may be more stable …”
Line 386:
“… The extension of EV-based method for drug prioritization should be pursued in the future.”
Line 424:
“… this EV-based workflow would be more appeal for clinical applications if …”
Line 428:
“… performance evaluation of MYCN-MV method for detecting MYCNamplification status of pediatric neuroblastoma in future.”
Conclusion;
Line 433:
“Future studies should investigate the EV-based method as an adjunct to the standard procedures …”